# Three-Dimensional (3D) Imaging Technology to Monitor Growth and Development of Holstein Heifers and Estimate Body Weight, a Preliminary Study

**DOI:** 10.3390/s22124635

**Published:** 2022-06-19

**Authors:** Yannick Le Cozler, Elodie Brachet, Laurianne Bourguignon, Laurent Delattre, Thibaut Luginbuhl, Philippe Faverdin

**Affiliations:** 1PEGASE, INRAE, Institut Agro, 35590 Saint-Gilles, France; philippe.faverdin@inrae.fr; 2Department of Animal Production, Agrifood, Nutrition (P3AN), Agro Rennes-Angers, 35042 Rennes, France; elodie.brachet@agrocampus-ouest.fr (E.B.); laurianne.bourguignon@agrocampus-ouest.fr (L.B.); 33D Ouest, 5 Rue de Broglie, 22300 Lannion, France; laurent.delattre@3douest.com (L.D.); thibault.luginbuhl@3douest.com (T.L.)

**Keywords:** 3D imaging, heifer, monitoring, traits, estimation

## Abstract

The choice of rearing strategy for dairy cows can have an effect on production yield, at least during the first lactation. For this reason, it is important to closely monitor the growth and development of young heifers. Unfortunately, current methods for evaluation can be costly, time-consuming, and dangerous because of the need to physically manipulate animals, and as a result, this type of monitoring is seldom performed on farms. One potential solution may be the use of tools based on three-dimensional (3D) imaging, which has been studied in adult cows but not yet in growing individuals. In this study, an imaging approach that was previously validated for adult cows was tested on a pilot population of five randomly selected growing Holstein heifers, from 5 weeks of age to the end of the first gestation. Once a month, all heifers were weighed and an individual 3D image was recorded. From these images, we estimated growth trends in morphological traits such as heart girth or withers height (188.1 ± 3.7 cm and 133.5 ± 6.0 cm on average at one year of age, respectively). From other traits, such as body surface area and volume (5.21 ± 0.32 m^2^ and 0.43 ± 0.05 m^3^ on average at one year of age, respectively), we estimated body weight based on volume (402.4 ± 37.5 kg at one year of age). Body weight estimates from images were on average 9.7% higher than values recorded by the weighing scale (366.8 ± 47.2 kg), but this difference varied with age (19.1% and 1.8% at 6 and 20 months of age, respectively). To increase accuracy, the predictive model developed for adult cows was adapted and completed with complementary data on young heifers. Using imaging data, it was also possible to analyze changes in the surface-to-volume ratio that occurred as body weight and age increased. In sum, 3D imaging technology is an easy-to-use tool for following the growth and management of heifers and should become increasingly accurate as more data are collected on this population.

## 1. Introduction

In dairy production, proper management of the non-productive period before first calving has important implications for the future production, reproduction, and/or longevity of dairy cows. The significance of this period is often underestimated, but rearing a heifer from birth to first calving (24 months of age) is costly: USD 1500–2000 in France [1] and USD 2241 in the USA [2]. In a 2001 study by Tozer and Heinrichs, the two management factors that most influenced the cost of raising replacement animals were the milking herd replacement rate and age at first calving (AFC). Correspondingly, decreasing the length of the non-productive period prior to first calving is one of the most effective ways to decrease rearing costs [3]. In Holstein cows, typical recommendations for AFC are around 24 months of age, although it is usually higher: in France, for example, it was around 30 months of age in 2019 [4]. For a 100-cow herd, Tozer and Heinrichs [5] estimated that a one-month reduction in AFC could lower the costs of a replacement program by USD 1400. There are thus long-standing recommendations for careful monitoring of developmental indicators, including changes in body weight (BW) and/or in morphological criteria such as heart girth (HG) or withers height (WH) [6]. Indeed, when this information is recorded on-farm, it allows farmers to closely follow the growth of their heifers, compare it to optimum objectives for BW or morphological traits, and, if necessary, adapt their management practices accordingly. BW is seldom recorded directly on farms, but equations have been proposed for the accurate prediction of BW from morphological traits [7,8]. BW is also used to adapt the dosage of pharmacological treatments to individual animals, although in some cases body surface area (BSA) is more appropriate [9]. BSA is difficult to determine directly but can be predicted from metabolic weight (BW^0.75^ or similar) based on equations first established by Elting [10]. When BSA is available, it is also possible to precisely calculate energy and protein maintenance requirements, since they are linked to both BW and morphological development, which can continue up to lactation 3 or higher [11].

Despite their utility and importance, however, growth indicators are rarely used on farms, mainly due to the amount of time required for data collection, the risks inherent in manipulating animals, and the stress involved for both animals and farmers. Indeed, traditional weighing systems require handling of animals, which involves risks of injuries for the farmer (loading of an animal, blowing of horns or feet, etc.) or to the animals (trapped or broken legs, for example). An attractive solution to this problem could be the use of technology that automatically and continuously records all traits without the need for direct contact with animals. Devices such as automatic weighing or devices based on a walk-over scale limit or even avoid these risks. However, in this case, only BW is usually available. In this context, three-dimensional (3D) imaging has demonstrated promise, in comparison to manual measurements [12,13]. It has been used to effectively measure morphological traits and estimate BSA and volume in adult dairy cows, and thus to estimate BW and its changes during lactation [11,12,13]. To date, however, this technology has only been used on adult dairy cows, and it is unclear to what extent the same approach can be applied to growing calves or heifers. To this end, we decided to conduct a pilot project that would evaluate the use of 3D imaging on a limited number of growing heifers (*n* = 5), for which it would be possible to obtain the frequent, but costly and time-consuming, records needed for testing and validation [12]. Using this test population, we aimed to evaluate if 3D imaging is of interest to closely monitor the growth and the development of dairy heifers and, thanks to equations we previously developed in adult cows, if it is possible to estimate their BW with this technology. This information could then be used to inform further developments in the use of this technology.

## 2. Materials and Methods

Five replacement Holstein heifers were randomly selected 5 weeks after birth. Their growth and development were monitored up to 20–21 months of age, close to the end of their first gestation. The sample was intentionally kept small for two reasons: first, because of practical considerations associated with the labor force and barn space available, and second, because the recording system (Morpho3D device, 3D Ouest, Lannion, France) had never been used on growing animals before, and these animals thus constituted a pilot population. Heifers were born and reared at the Mejusseaume experimental station of INRAE, Dairy Nutrition and Physiology unit (IE PL, 35,560 Le Rheu, France) (https://doi.org/10.15454/yk9q-pf68, accessed on 7 June 2022). They were reared using routine farm procedures according to French animal-welfare regulations, and no specific authorization regarding ethical considerations was required.

### 2.1. General Animal Management

At the IE PL, the calving season starts in September and ends in mid-January. The heifers selected for this project were born between the 27 of September and the 13 of November 2017. They were first inseminated during the second winter of rearing, after the detection of natural heat, so that first calving occurred at around 24 months of age. During the initial phase of rearing (0–10 days), calves were fed colostrum and non-commercialized full-fat milk. From 10 days of age until weaning, a reconstituted milk replacer (135 g milk powder + 865 g water per L; 23.9% crude protein and 19.0% fat) was offered. Heifers were group-housed indoors on deep straw bedding and fed individually with an automatic milk feeding system, with free access to fresh water, straw, and hay until weaning. The milk amount regularly increased until 21 days of age (8.2 kg of reconstituted milk replacer) and remained constant until 63 days of age. It then decreased to 2 kg/day/heifer until weaning (77 days on average). As a supplement to the milk, all calves were fed ad libitum with a complementary mixed ration (CMR1) (Table 1).

From weaning until 4 months of age, CMR1, containing 47.5% of concentrate 1, was offered ad libitum until the amount of concentrate 1 reached 2 kg dry matter (DM)/head/day. CMR1 was then capped at this level. From 4 to 8 months of age, a total mixed ration (TMR1), containing 20% of concentrate 1, was distributed ad libitum until the level of concentrate 1 reached 2 kg DM/head/day (i.e., a maximum of 10 kg TMR1/head/day). During this period and until they were turned out to pasture (when the youngest heifer reached 183 days of age, i.e., 6 months of age on average), heifers remained group-housed on deep straw bedding with ad libitum access to fresh water and straw.

Beginning in mid-May 2018, heifers (183 to 230 days of age) were turned out to pasture and rotationally grazed on a perennial ryegrass. After a 5-day transition phase, they received a supplement of 1 kg DM/head/day of concentrate 2 (Table 1) throughout the remainder of the grazing season. Because grass availability and/or quality were insufficient to maintain the desired growth rates during summer (heat stress), the heifers received up to 2.5 kg DM/head/day of additional CMR2, plus 1 kg DM/head/day of concentrate 2. At the end of the grazing season (beginning of November), heifers were group-housed on deep straw bedding and received 3.8 kg DM/head/day of TMR2, plus 1 kg of concentrate 2 (Table 1). They had free access to fresh water, straw, and mineral supplements. Heifers were then inseminated on the first detected natural heat, starting December 1st. They were turned back out to pasture in the spring (March 2019) and received no additional feed except for grass, along with supplemental vitamins and minerals. Data recording for this experiment stopped 2 months (on average) before the expected start of the calving season. All diets (Table 1) were formulated for each stage of growth, according to the recommendations and procedures of Agabriel and Meschy [14], with the goal of reaching a targeted average daily gain per period, based on the initial BW and feeding treatment (see also [15] for more details). Health and treatment information was recorded for all heifers, but no problems occurred during the course of the experiment.

### 2.2. Measurements

Heifers were weighed every 14 days from birth to weaning, every 21 days from weaning until turning out to pasture (183 to 230 days of age), and every 28 days until the end of the experiment. Weighing usually occurred between 10:00 and 12:00 to keep conditions similar. Between weighing dates, BW was interpolated in order to have data for all heifers at similar ages. Daily gains were then calculated for each animal. 

Each heifer passed through the Morpho3D device (3D Ouest, Lannion, France) once a month, from November 2017 to August 2019, so that 3D images could be captured. Image acquisition was rapid when animals were getting used to the device and took on average 6 s. The dates of image capture usually corresponded to the weighing dates, and if not, BW was interpolated as described above. During image acquisition, animals were standing still. During the initial passages, heifers were held firmly by a technician or, in some cases, restrained in a feed fence while images were acquired. As they grew accustomed to the device, it was no longer necessary to restrain them. Volume and surface area were estimated from either the partial or the total volume and surface area captured, as described by Le Cozler et al. [13] 

Briefly, the Morpho3D device is a sliding acquisition system, located near the weighing station, that has five cameras on the sides and top of the portal. Each camera is paired with a laser projector. While the portal moves from back to front, each camera takes 80 images per second, yielding a total of 2000 images. Images of the laser stripes projected onto the heifer are captured by the corresponding camera and sent to a computer. After processing, a 3D reconstruction of the heifer is generated, i.e., a single point cloud representation of the entire heifer (see [12] for additional details about Morpho3D). Finally, surface normals are estimated from the point cloud, and a screened Poisson surface reconstruction algorithm is applied to build a triangulated mesh [16] using Meshlab open-source software [17]. Metrux2α software (3D Ouest, Lannion, France) was used to perform linear measurements and estimate morphological traits, BSA, and volume [12].

For 3D images, the quality of the reconstruction has a strong influence on the measurements that can be obtained. In some cases, the main body of the heifer was digitized properly but its head or legs were not, meaning that the animal had moved slightly during image capture. This distorted its BSA and volume and resulted in the main distortion problem we observed: so-called “skirts” on the legs (Figure 1). 

This occurred more often with the youngest calves (less than 6 months of age), probably because, in addition to movement, the system was occasionally unable to record enough points for very small legs. The reconstruction software then filled in the empty area between points located on different legs, creating the so-called “skirt”. We therefore scored each image on a scale from 1–4 (1: not used; 2: partially usable; 3: some defects; 4: sufficient) to reflect the degree to which it could be used to determine BSA and/or volume (Table 2). 

For some poor-quality images, certain morphological traits (e.g., WH) could still be measured, even if BSA and/or volume could not be estimated correctly. However, as we took several 3D images of each heifer each month, it was generally possible to obtain all the information needed. Technical problems in December 2018 (i.e., at 475 days of age on average) prevented images from being recorded that month. Overall, each heifer was weighed at least 20 times, and a total of 101 images was taken. Of these, 69 were of sufficient quality for further analysis, with a minimum of 13 and a maximum of 17 images per heifer.

### 2.3. Analysis

Metrux2α software (3D Ouest, Lannion, France) was used to quantify morphological traits such as heart girth (HG), chest depth (CD), hip width (HW), and buttock width (KW) from the 3D images (see [12] for details). The validity (repeatability and reproducibility) of this method of measurement had been verified by earlier work in which we compared imaging data to “gold standard” manual measurements [12]. We therefore performed no manual measurements on the animals in this study. The selected morphological traits were chosen because farmers can perform them on-farm, using rule-tape for example. 

Body surface area (BSA) and/or volume determined from the Morpho3D device were used to estimate BW using the equations of Elting [10] and Le Cozler et al. [13]. Even though more recent authors have published updated versions (described by Le Cozler et al. [13]), the equations of Elting [10] are still commonly used, probably because BSA remains difficult to routinely determine. Elting [10] published equations for adult cows Equation (1) and growing heifers Equation (2). More recently, our group published equations to estimate BW as a function of BSA Equation (3) or other morphological traits Equation (4), but only for adult cows [13]. In that same study, we also created an equation to calculate BW based only on volume Equation (5) [13]. Here, we used these equations to calculate body weight for each animal, and these values were then compared to measurements recorded directly from animals.
BSA = 0.0839 × BW^0.67^(1)
BSA = 0.147 × BW^0.56^(2)
BW = 102.3 × BSA − 30.3(3)
BW = 812.1 × volume − 81.4 × BSA + 343.8 × KW+ 273.8 × HW + 208.8 × HG + 113.7 × WH − 280.7(4)
BW = 827.5 × volume + 45.8(5)

Data were first visualized using the *ggplot2* package (v. 3.2.1) [18] of R software [19] and the difference between estimated and recorded values was plotted. We also used a linear regression model based on the Car procedure in R software to compare estimated and recorded values of BW. Then, we re-estimated the coefficients used in equations 3, 4, and 5 for young animals, using an approach similar to that described in [13]. However, because of the limited number of animals and observations in this study, it was not possible to split our data into testing and validation sets; thus, no validation could be performed (only estimation).

## 3. Results

Growth curves were similar for all heifers except one (No. 7247), which grew more quickly than the others, especially after 150 kg BW (Figure 2a). Careful examination revealed nothing abnormal, however, so heifer 7247 was kept in the analysis.

At 6 and 12 months of age, heifers reached mean BW values of 227.5 kg and 366.8 kg, with standard deviations of 49.5 and 47.2 kg, respectively. They weighed 383.8 (±20.8) kg at around 15 months of age, when first inseminated. Heart girth (Figure 2b) and WH (Figure 2c) increased according to asymptotic curves: at 6, 12, and 15 months of age (Table 3), HG equaled 151.7, 188.1, and 201.8 cm, respectively, while the corresponding WH values were 116.7, 133.5, and 140.5 cm, respectively. 

There was a surprising amount of variation in HG after 500 days of age, probably because the heifers were fed only with grass of varying quality and density. The volume of heifers increased linearly with age (Figure 2d), from 0.28 to 0.49 m^3^ at 6 and 15 months of age, respectively (Table 3), while BSA (Figure 2e) increased asymptotically (4.16 to 5.42 m^2^ at 6 and 15 months of age, respectively; Table 3). The faster increase in volume relative to BSA, according to BW changes, resulted in a surface-to-volume ratio that decreased exponentially (Figure 2f). At 20 months of age, BW was 6.5 times larger than that at 2 months of age, whereas volume was 5.1 times larger (0.12 to 0.61 m^3^) and BSA only 2.3 (2.73 to 6.37 m^2^).

Based on the traits determined from 3D images, it was possible to estimate BW and compare these values to recorded BW (Figure 3). The comparison with the models from Elting [10] is presented on Figure 3a–b, and comparison with the models we previously determined on adult cows is presented on Figure 3c–e.

When regression models were forced to pass through 0, Equations (1)–(3) and (5) overestimated BW, and even the most accurate equation, Equation (4), slightly underestimated it. 

Measured and estimated (Equation (4)) values of body weight at 20 months of age were equal to 551 (±71.8) and 561 (±70.0) kg, respectively. At 12 months of age, these values were 367 (±45) and 400 (±36 kg), respectively, while at 6 months of age, they were 202 (±43.4) and 250 (±43.5) kg, respectively. This corresponded to overestimations of +1.8%, +8.2%, and +19.2%, respectively. When models were not forced to pass through zero, they usually had lower R^2^ values. A re-consideration of the coefficients used in Equations (3)–(5) for young animals resulted in new equations as follows:BW = 118.2 × BSA − 259.3 (R^2^ = 0.93)(6)
BW = 728.8 × volume − 0.36 × HW − 91.3 (R^2^ = 0.98)(7)
BW = 922.3 × volume − 34.4 (R^2^ = 0.97)(8)

In the complete equation (Equation (3)), only volume and HW were significant, with no effect of BSA, KW, HG, or WH.

## 4. Discussion

Our results reveal that it is possible to estimate body weight in this population, based on volume and/or morphological traits. The strong correlation observed here between volume and body weight (R^2^ = 0.97) supports the findings of Costigan et al. [8], who concluded that volume is the most suitable predictor of live weight. Similarly, previous work by our group on adult cows found that body weight predictions based on volume were highly accurate (RMSE of 24.5 kg for animals weighing between 550 and 800 kg; Le Cozler et al. [13]). Here, our results were also consistent with those of Costigan et al. [8] in that BW and volume increased in a similar way from 2 to 20 months of age (×6.5 and ×5.1, respectively), while the increase in BSA was lower (×2.3). In adult cows, BW prediction was slightly improved when morphological traits and BSA were considered together with volume (R^2^ = 0.98), but in the present study, only hip width together with volume had a significant effect. More data are still needed to determine the most accurate equation. However, even though the equation based only on volume was not the most precise, it may ultimately hold the most promise for practical applications: volume can be easily and automatically calculated from 3D images, while the inclusion of other variables requires manual measurements that are time-consuming and potentially dangerous. In any case, improved automation (e.g., image preparation, measurement) is required for this tool to reach its potential for routine on-farm use. 

Over- or underestimations in traits obtained from 3D images may have resulted in over- or underestimations in BW. The low quality of some images and the difficulty in measuring some of the traits used in the equations may also have been partially responsible for these errors. For certain distortion problems, we hypothesized that the heifer’s hair decreased image quality; we therefore tested different preparation (shorn or not) and presentation (on a raised platform or not) of the young calves, but these had no effect on image quality. Another potential explanation is that the paired cameras and laser projectors were located too far from the calves. Therefore, it may be necessary to adapt the device to the smaller size of calves to increase image quality. The presence of skirts can be explained by the fewer points acquired in point clouds from the smaller limbs of young calves, but they may have also occurred because Morpho3D projects lasers in a single plane. One possible solution may be to project acquisition planes that cross at different angles. 

This method for estimating surface-to-volume ratio could be useful in studying animals’ ability to adapt to changes in temperature, because heat losses decrease as the ratio decreases. The surface-to-volume ratio is high in young animals and decreases as they grow, meaning that calves lose heat quickly. As a complement to this initial analysis, we supplemented the data from the heifers with data from adult cows, which were monitored in fall–winter 2018 and 2019 (Figure 4, black circles). 

The combined data indicated that heavy cows have a lower surface-to-volume ratio compared to young animals. Regulating body temperature is then probably different between these animals, and this underlines the importance of breeding conditions for young animals.

Ultimately, one of the main uses of BSA is to estimate the maintenance requirements of dairy cows. The model used most often to estimate BSA was developed by Mitchell [20], although similar models have been published by other authors [10,21,22]. We previously compared these models for adult cows [13], and the models of Elting [10] and Mitchell [20] provided results that were similar to ours. Because image reconstruction or low-quality images from the Morpho3D device may yield errors, our method may sometimes over- or underestimate the true BSA. However, other methods that estimate BW from BSA are also limited since they can usually be applied only to a few types of calm, easily handled animals. In comparison, the Morpho3D device can acquire images from many types of animals—such as cross-bred bulls (Swiss Brown × Limousin), Swiss horses, and donkeys—to calculate their BSA (5.27, 6.46, and 4.41 m^2^, respectively; C. Xavier, pers. comm.) and, from that, their BW. Finally, using the Morpho3D device, Xavier et al. [23] concluded that 3D imaging technology appears to be of great interest for estimating the chemical composition of living animals.

## 5. Conclusions

From the results of the present study, it is clear that the analysis of 3D images of entire animals is a promising technology for estimating indicators of growth and development in dairy heifers, which are of interest in dairy production, and probably also in other types of animal production. Because our results are limited to five heifers, though, they need to be confirmed in a larger group of animals. These findings, although preliminary, indicate that regular 3D image acquisition can be used to monitor and study the dynamics of morphological trait development during the growth of heifers. This approach can also be used to estimate traits of interest such as BSA and volume, which are currently difficult to determine. Using these data, it is then possible to determine BW directly from volume with a high degree of accuracy without special contact or handling of animals, which could be of particular interest in beef or bull production.

## Figures and Tables

**Figure 1 sensors-22-04635-f001:**
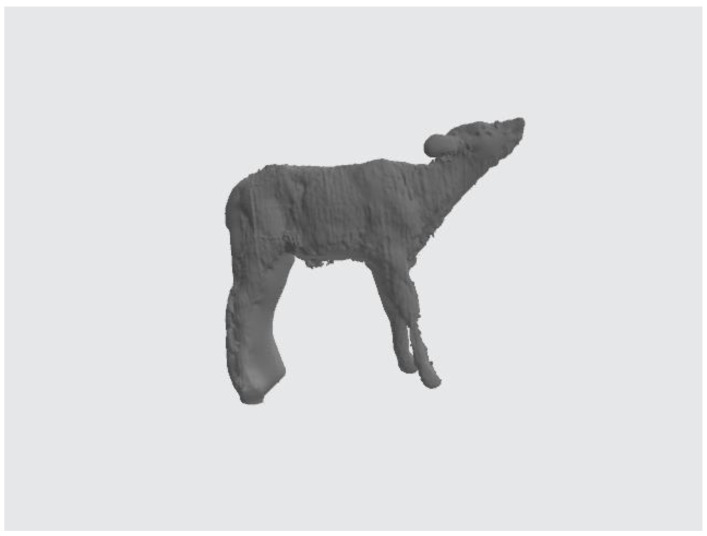
Abnormal image representing the presence of a “skirt” on the legs. When the number of points is insufficient, the software fills in the volume between the points located on different legs (which is not the case when there are enough points).

**Figure 2 sensors-22-04635-f002:**
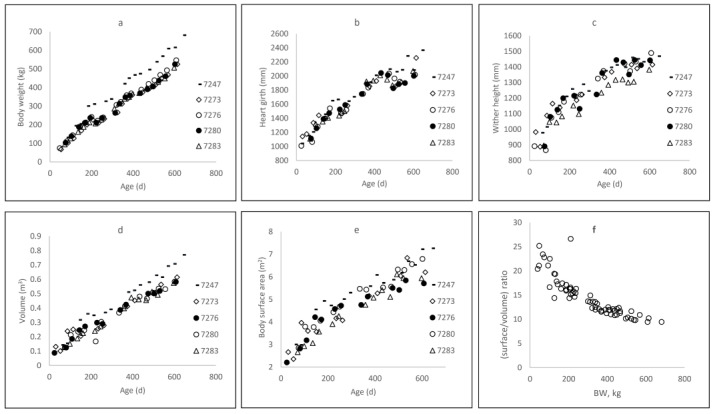
Dynamics of body weight (**a**), heart girth (**b**), wither height (**c**), surface area (**d**), and volume (**e**) as a function of age, and (**f**) surface-to-volume ratio as a function of body weight for five Holstein heifers, from 5 weeks of age until the end of gestation.

**Figure 3 sensors-22-04635-f003:**
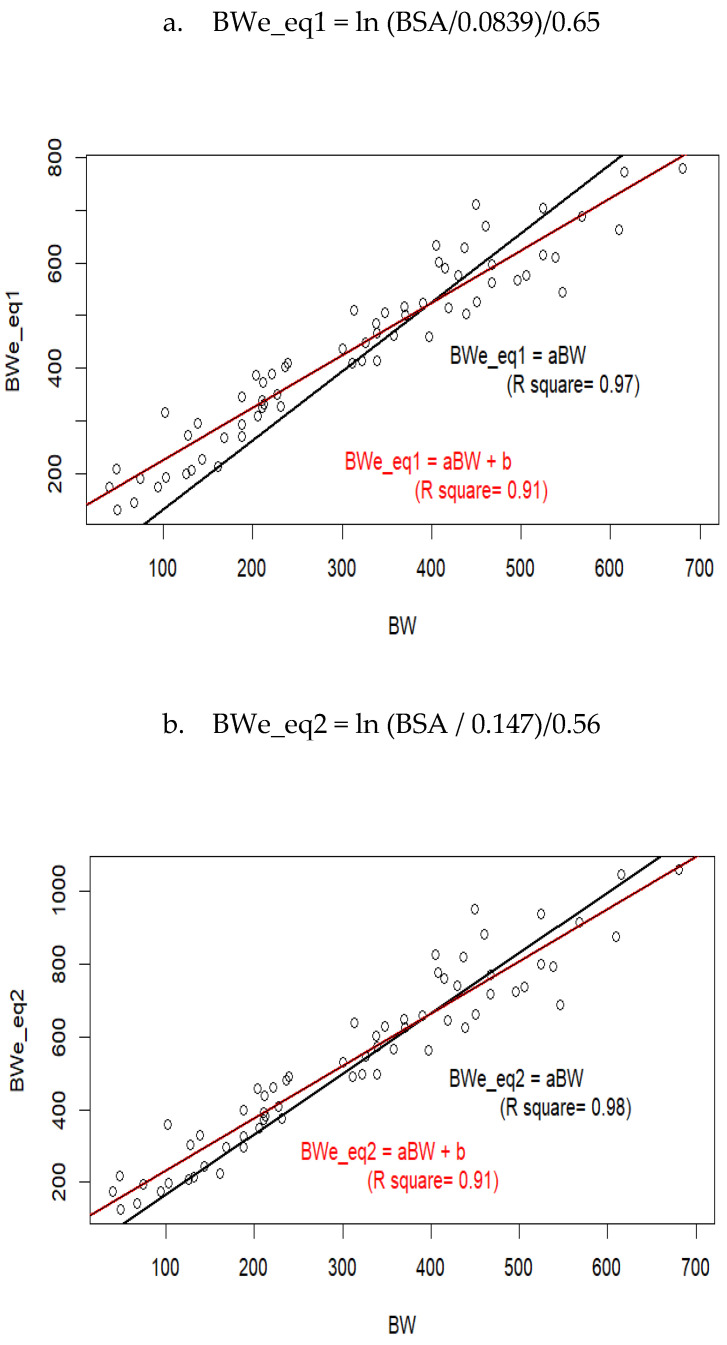
Measured values of body weight (BW) compared to values estimated (BWe) as a function of body surface area (BSA), volume, withers height (WH), and/or buttock width (WH), using equations from Elting [10] (**a**,**b**); or Le Cozler et al. [13], (**c**–**e**).

**Figure 4 sensors-22-04635-f004:**
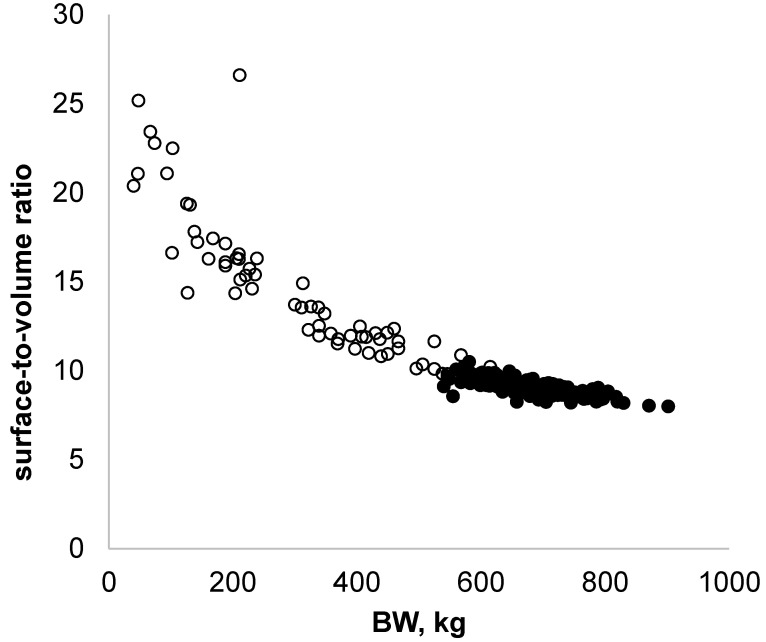
Changes in surface-to-volume ratio as a function of body weight for five Holstein heifers, from 5 weeks of age until the end of gestation (empty circles; *n* = 68 observations), supplemented with data from adult cows (full circles; *n* = 177 observations).

**Table 1 sensors-22-04635-t001:** Ingredients and chemical composition of the complementary mixed ration (CMR) and total mixed ration (TMR) used during the rearing period.

Item ^1^ Stage of Growth, Age in Months	CMR1 (0 to 2–4)	TMR1 (4 to 6–8)	CMR2 (9 to 11)	TMR2 (11 to 15)
Ingredients (%, unless noted)				
Maize silage	47.5	72.0	80.0	79.0
Soybean meal	-	8.0	20.0	21.0
18% CP alfalfa pellets	5.0	-	-	-
Urea		-	-	-
Vitamins and minerals	-	-	-	-
Concentrate 1 ^2^	47.5	20.0	-	
Concentrate 2 ^3^ (kg/head/day)	-	-	1.0	1.0
Estimated chemical composition				
DM (%)	51.4	42.0	42.2	42.1
PDIE (g/kg DM)	93.0	93.1	104.5	106.2
PDIN (g/kg DM)	79.8	84.0	108.7	111.3
UFL (g/kg DM)	0.96	0.96	0.98	0.99

^1^ Abbreviations: CP: crude protein; DM: dry matter; PDIE: protein digestible in the small intestine, g/kg; PDIN: protein digestible in the small intestine, g/kg; UFL: forage unit for lactation, g/kg. ^2^ Chemical composition: 88.7% DM, 118 g PDIE, 114 g PDIN, 1.05 UFL. ^3^ Chemical composition: 87.9% DM, 81 g PDIE, 90 g PDIN, 0.96 UFL.

**Table 2 sensors-22-04635-t002:** Scoring scale from 1–4 (1: not used; 2: partially usable; 3: some defects; 4: sufficient) of 3D images, according to the age of heifers.

Note	1	2	3	4
Animal aged less than 6 mo	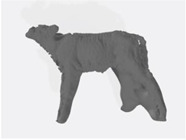	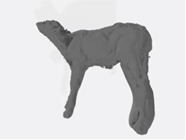		
Animal aged 6 mo or more	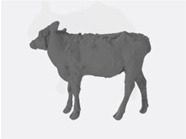	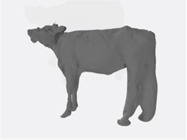	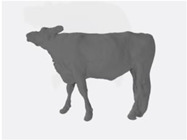	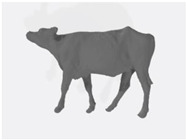

**Table 3 sensors-22-04635-t003:** Mean and standard deviation (in brackets) of selected morphological traits, surface area, and volume estimated from 3D images of five growing heifers at different stages of growth. BW was recorded using a weighing scale.

Stage of Growth	Age, days	BW, kg	BSA, m^2^	Volume, m^3^	HW, mm	CD, mm	HG, mm	KW, mm
2 months	62.0 (11.8)	84.5 (16.8)	2.73 (0.24)	0.12 (0.01)	903.0 (42.9)	423.8 (13.9)	1122.3 (43.5)	205.5 (13.5)
4 months	136.4 (13.0)	173.6 (30.3)	3.84 (0.57)	0.25 (0.05)	1118.8 (47.3)	543.1 (19.9)	1406.6 (41.5)	283.4 (27.7)
6 months	171.0 (9.6)	227.5 (49.5)	4.16 (0.56)	0.28 (0.06)	1167.1 (58.3)	585.0 (31.0)	1517.3 (104.3)	299.5 (15.3)
8 months	246.8 (8.8)	228.0 (13.0)	4.45 (0.29)	0.28 (0.02)	1159.1 (55.7)	588.9 (14.7)	1518.5 (51.3)	343.1 (15.4)
12 months	369.2 (8.6)	366.8 (47.2)	5.21 (0.32)	0.43 (0.05)	1335.3 (58.9)	718.1 (27.3)	1881.2 (37.1)	403.2 (28.9)
15 months	455.8 (19.3)	393.8 (20.6)	5.48 (0.23)	0.49 (0.04)	1405.1 (49.2)	743.1 (45.4)	2018.1 (42.5)	424.6 (23.1)
20 months	604.8 (6.3)	543.4 (42.5)	6.37 (0.62)	0.61 (0.06)	1431.1 (39.4)	786.1 (17.9)	2085.5 (101.6)	508.6 (11.8)

Signification: HG: heart girth; CD: chest depth; HW: hip width; KW: buttock width; BSA: body surface area (BSA); BW: body weight.

## Data Availability

Not applicable.

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
