# Peer review of "Three-Dimensional (3D) Imaging Technology to Monitor Growth and Development of Holstein Heifers and Estimate Body Weight, a Preliminary Study"

_sensors, 2022, doi:10.3390/s22124635_

Round 1

Reviewer 1 Report

This study evaluated the use of 3D imaging to estimate body weight and other physical traits in heifers from birth to maturity. Although this was a pilot study using only 5 animals, the paper in novel in testing the adequacy in growing heifers. The technology showed very good consistency with direct measurements. The paper is well presented and clearly written. 

The authors might explain how the use of this technology is safer than weighing the animals, as presumably they have to 'pass through the device'. How is this different to walking them through a race with walk-over scales?

While there are many other useful applications for the use of imaging, since it can easily capture many measurements in comparison to simply weighing animals, I am not certain this is justified for commercial farms excluding studs which might want to select on various traits. I am assuming it requires animals to walk into a device, which is similar effort as weighing animals.  I would have thought that weight is the key trait to monitor growth, and weights (relevant to breed) are key targets for different stages of growth. Weight, however, does not indicate the degree of fatness, with condition scores being as important as weight in attaining adequacy for mating. This is particularly relevant when the frame size of animals varies.  Why was condition score not also recorded? The cost of the device, rather than scales, is not mentioned, but I am assuming would not be competitive.

Minor comments:

line 96-97, 115 - font size changes?

line 99 - state manufacturer, place

line 155 - How long does the cow have to stand in the device?

line 202 - delete determine (retain quantify)

Figure 2 - it would be preferable to describe a-c, rather than lump as morphological traits in the title

Results - how much of the variability might be accounted for by gut fill, since unfasted weights do not seem to have been used?

Figure 4 - Vertical axis title should not be in brackets

line 320-322 - delete or revise - you cannot assume there is a heat stress impact as this was not measured, and supporting references were not provided.

Author Response

Reviewer 1 :

Comments and Suggestions for Authors

This study evaluated the use of 3D imaging to estimate body weight and other physical traits in heifers from birth to maturity. Although this was a pilot study using only 5 animals, the paper in novel in testing the adequacy in growing heifers. The technology showed very good consistency with direct measurements. The paper is well presented and clearly written. 

AU: thank you for the remark. We agree, it is a preliminary work that is still going on.

The authors might explain how the use of this technology is safer than weighing the animals, as presumably they have to 'pass through the device'. How is this different to walking them through a race with walk-over scales?

AU: this is correct. This only is true if you compare to traditional weighing system, since in this case, you need to handle animals. It is not necessary with present device. The walk-over scales is an alternative option. We tested it on the past, but we were not fully satisfied because of frequent errors and precision. We rewrote , lines 66-73

“ Indeed, traditional weighing systems require handling of animals, which involves risks of injuries for the farmer (loading of an animal, blowing of horns or feet, etc.) or to the animals (trapped or broken legs, for example). An attractive solution to this problem could be the use of technology that automatically and continuously records all traits without the need for direct contact with animals. Devices such as automatic weighing or based on walk-over scale limit or even avoid these risks. However, in this case, only BW is usually available.”

While there are many other useful applications for the use of imaging, since it can easily capture many measurements in comparison to simply weighing animals, I am not certain this is justified for commercial farms excluding studs which might want to select on various traits. I am assuming it requires animals to walk into a device, which is similar effort as weighing animals.  I would have thought that weight is the key trait to monitor growth, and weights (relevant to breed) are key targets for different stages of growth. Weight, however, does not indicate the degree of fatness, with condition scores being as important as weight in attaining adequacy for mating. This is particularly relevant when the frame size of animals varies.  Why was condition score not also recorded? The cost of the device, rather than scales, is not mentioned, but I am assuming would not be competitive.

AU: we definitively agree. This first device helped us to develop precise algorithms, but also a new device that we just tested, more suitable for on-farm use, but for this purpose, we will used information from the second one to develop this third and hopefully, final one. We are progressing step-by-step, trying to be as efficient as possible (this is why also we did not want to use too much animals at start). We agree it is not competitive yet if you are focusing only on BW, but if you are interested in other traits such as volume or surface, this equipment is of great interest. The cost of the equipment itself is not very important. It is of course the mathematical software and algorithms we developed that took time and resulted in a high costs. The final device will able to estimate BW, morphological traits, surface.... but probably, also, able to predict chemical composition. The Morpho3D device already gives such information, but is not suitable for on-farm use.

Minor comments:

line 96-97, 115 - font size changes?

AU: We change the font

line 99 - state manufacturer, place

AU: We added line 97 : “Morpho3D device; 3D Ouest, Lannion, France)” (se hope we understood correctly the question ?)

line 155 - How long does the cow have to stand in the device?

AU: We added this information (line 158-160): “Image acquisition was rapid when animals were getting used to the device and took on average 6 seconds. “

Line 202 - delete determine (retain quantify)

AU : done line 213

Figure 2 - it would be preferable to describe a-c, rather than lump as morphological traits in the title

AU: lines 246-247,  we re-wrote : “Figure 2. Dynamics of body weight (a), heart girth (b), wither height (c), surface (d) area, and volume (e)…”

Results - how much of the variability might be accounted for by gut fill, since unfasted weights do not seem to have been used?

Au: this is an interesting and important question. We discuss a lot about this and it is very hard to use unfasten weights in such animals, that are outside while grazing. In addition, we estimated that it requires probably more than 18 hours for fasting, but it sometimes requires more than 24 h in gestating heifers. For a welfare point of view, we decided to work only with unfasted animals. In lactating adults cows, we sometimes saw difference greater than 80 kg when weighing at 8.00 AM and 6.00 PM for ex.

The commonly used double weighing would have been more suitable but it required time.

Figure 4 - Vertical axis title should not be in brackets

AU: changes have been performed (line 346)

line 320-322 - delete or revise - you cannot assume there is a heat stress impact as this was not measured, and supporting references were not provided.

AU: we changes to (lines 352-355): “ The combined data indicated that heavy cows have a lower surface-to-volume ratios compared to young animals. Regulating body temperature is then probably different between these animals, and underlines the importance of breeding conditions for young animals.”

Reviewer 2 Report

The manuscript "Three-dimensional (3D) imaging technology to monitor growth 2 and development of Holstein heifers and estimate body 3 weight, a preliminary study" enters the topics of the journal. In this study, the utility is verified. The analysis of 3D images of entire animals is a promising technology for estimating indicators of interest in animal production. This research was limited to five heifers and would need to be confirmed in a larger group of animals. Using 3D images is possible to determine the growth and other parameters of interest.

Weaknesses:

According to the articles seen, the 3d system is promising, but the work does not provide comparison with other methods, etc. It would be necessary to clearly highlight the scientific novelty of the study, since on the other hand the limited number of animals used limits the conclusions.

The technology, both live and after slaughter, has been used in species for slaughter. The sample used was very scarce and also simple parameters such as the height at the withers or the thoracic perimeter are not measured (which is not difficult to do). In cattle, the height of the rump is also usually measured as an indicator of growth. In addition, in bovines, the height at the hip is more reliable than the height at the withers. Why it is not done?.

Modeling of growth functions applied to animal production were not used (Gompertz, Logistics, Von Bertalanffy, Brody and Richards). I suggest to redo the data with both types of functions and compare them (3d measurement vs manual measurement). Perhaps statistically the small number of animals can be supplied.

As the number of animals is very small and they do not adjust to normal distributions, we have other problems. How have you considered them?

Abstract:

“Unfortunately, current methods for evaluation can be costly, time-consuming, and dangerous because of the need to physical manipulate animals, and as a result, this type of monitoring is seldom performed on farms.”

I think:

The main problem is the cost of handling, which is why very few weighings are made (frequently one at the beginning and another at the end of the phase). It would be convenient to have precise data in shorter intervals. The IoT can be of great help in this regard

Line 64:

I think: Despite their utility and importance, however, growth indicators are rarely used on farms, mainly due to the amount of time required for manipulating animals and breaking the routine of exploitation

Objective:

The objective, the title and the conclusions must coincide in their approach. I suggest you check it out

Figure 3 is not seen clearly enough.ble to cows with more lactations and other groups of animals?

It is a good job and I encourage its correction. It will improve it and facilitate its use of this type of alternative measurement inn animal production.

It is a good job and I encourage its correction. It will improve it and facilitate its use of this type of alternative measurement in animal production. This methodology will allow us to routinely work with large volumes of data (more frequent data per animal and greater volume of animals) and I think it is very promising.

Other bibliography than I recommend you

Validation of an Automated Body Condition Scoring System Using 3D Imaging

Automatic estimation of body weight and body condition score in dairy cows using 3D imaging technique

Using 3D Imaging and Machine Learning to Predict Liveweight and Carcass Characteristics of Live Finishing Beef Cattle 

High-precision scanning system for complete 3D cow body shape imaging and analysis of morphological traits

Three-Dimensional Imaging System of Dairy Cow Based on Virtual  Instrument

Author Response

Reviewer 2

Comments and Suggestions for Authors

The manuscript "Three-dimensional (3D) imaging technology to monitor growth 2 and development of Holstein heifers and estimate body 3 weight, a preliminary study" enters the topics of the journal. In this study, the utility is verified. The analysis of 3D images of entire animals is a promising technology for estimating indicators of interest in animal production. This research was limited to five heifers and would need to be confirmed in a larger group of animals. Using 3D images is possible to determine the growth and other parameters of interest.

Weaknesses:

According to the articles seen, the 3d system is promising, but the work does not provide comparison with other methods, etc. It would be necessary to clearly highlight the scientific novelty of the study, since on the other hand the limited number of animals used limits the conclusions.

AU: Yes, we agree and thank you for this remark. In a previous paper, we discuss and presented advantages and limits of these technology to manual measurements, considered as gold standard. And we compared also to some other imaging based technology. We added then (lines 73-74=: “…in comparison to manual measurements (Le Cozler et al., 2019a; 2019 b))

The technology, both live and after slaughter, has been used in species for slaughter. The sample used was very scarce and also simple parameters such as the height at the withers or the thoracic perimeter are not measured (which is not difficult to do). In cattle, the height of the rump is also usually measured as an indicator of growth. In addition, in bovines, the height at the hip is more reliable than the height at the withers. Why it is not done?

AU: In fact, we have performed most of these analysis but we decided to focus on some of them. The idea was here to give the proof of concept: was the system working and ? The morphological traits selected were the same as previously published and validated by on-farms users, at least in France. We presented the height at the withers and thoracic perimeter since these are commonly used in dairy cattle management (Heinrichs and Hargrove;, 1987; Le Cozler et al., 2019a; Xavier et al., 2022). But others could have been included; of course.

We also added (lines 368-370): “ And finally, using Morpho3D device, Xavier et al. (2022b) concluded that 3D imaging technology appears to be of great interest to estimate chemical composition from living animals.”

And the reference (lines 443-445)

Xavier C., Driesen C., Siegenthaler R., Dohme-Meier F., Le Cozler Y., Lerch S., 2022. Estimation of Empty Body and Carcass Chemical Composition of Lactating and Growing Cattle: Comparison of Imaging, Adipose Cellularity, and Rib Dissection Methods. Translational Animal Science (in press)

Modeling of growth functions applied to animal production were not used (Gompertz, Logistics, Von Bertalanffy, Brody and Richards). I suggest to redo the data with both types of functions and compare them (3d measurement vs manual measurement). Perhaps statistically the small number of animals can be supplied.

AU: we agree and in fact we also studied these different approaches in a previous work (Sauder et al., 2012). In this study, we worked on modelling growth curves of animals, which are commonly based on modelling growth curves. Brown et al. (1976) compared different mathematical functions (logistic, Gompertz, von Bertalanffy, Brody and Richards functions) for representing body weight (BW) over time in different cattle breeds. These parametric models can fit average levels well, but ignore individual variations.  We first tried these approaches too; but because we had too few data (and an extreme animal), it was not possible to conclude statistically. In addition, thee previously cited models may not be fully relevant for heifers, because they require the asymptotic value of BW, i.e. adult BW at lactation 2 or 3 (not available here). Non-parametric modelling, such as approaches based on spline functions, could have been used was used, like in Sauder et al; but we though that this was not the purpose of present paper.

As the number of animals is very small and they do not adjust to normal distributions, we have other problems. How have you considered them?

AU:  we agree, but we did not took into account the distribution in the present study. That will be an important point to consider in future.

 Abstract:

“Unfortunately, current methods for evaluation can be costly, time-consuming, and dangerous because of the need to physical manipulate animals, and as a result, this type of monitoring is seldom performed on farms.”

I think:

The main problem is the cost of handling, which is why very few weighings are made (frequently one at the beginning and another at the end of the phase). It would be convenient to have precise data in shorter intervals. The IoT can be of great help in this regard

AU: yes, we agree. Developing a tool that can give us frequent and regular information is important. For example, our second device is based on on-shot technology, that allows to have full image, without handling. For its validation, 13 heifers passed through it 5 times the day of measurement, every week.

Line 64:

I think: Despite their utility and importance, however, growth indicators are rarely used on farms, mainly due to the amount of time required for manipulating animals and breaking the routine of exploitation

AU: as a scientist involved in heifers’ management, I can only (unfortunately) agree and it is a pity. Farmers seldom realize what is the real cost of rearing and the real benefit to closely monitor growth. But in the future; with automatic and cheap device, this could (let’s hope so !) change…

Objective:

The objective, the title and the conclusions must coincide in their approach. I suggest you check it out

AU: To coincide with the title, we re-wrote:

Lines 85-88: “Using this test population, we aimed to evaluate if 3D imaging is of interest to closely monitor the growth and the development of dairy heifers and thanks to equations we previously developed in adult cows, if it is possible to estimate their BW with this technology.

And also:

Lines 374-375: “…animals is a promising technology for estimating indicators on growth and development in dairy heifers, which are of interest in dairy production, and probably also in other…”

Figure 3 is not seen clearly enough.

AU: we decided to split this figure into 2 new ones:

Line 275 “Figure 3. Measured values of body weight (BW) compared to values estimated (BWe) as a function of body surface area (BSA), from Elting (1926)”

Line 286 ‘Figure 4. Measured values of body weight (BW) compared to values estimated (BWe) as a function of body surface area (BSA), volume, withers height (WH), hip width (HW), and/or buttock width (KW), from Le Cozler et al. (2019b).’

We decided not to keep the difference between prediction and observation, as presented previously on figure 3.

Figure 4 in previous version is now figure 5

It is easier to read and we hope clearer now.

It is a good job and I encourage its correction. It will improve it and facilitate its use of this type of alternative measurement in animal production. This methodology will allow us to routinely work with large volumes of data (more frequent data per animal and greater volume of animals) and I think it is very promising.

AU: thanks again for the remark and the encouragement. We also agree this final remark and we will try to do our best in the future and get this technology / approaches available for all interested users.

Other bibliography than I recommend you

Validation of an Automated Body Condition Scoring System Using 3D Imaging

Automatic estimation of body weight and body condition score in dairy cows using 3D imaging technique

Using 3D Imaging and Machine Learning to Predict Liveweight and Carcass Characteristics of Live Finishing Beef Cattle 

High-precision scanning system for complete 3D cow body shape imaging and analysis of morphological traits

Three-Dimensional Imaging System of Dairy Cow Based on Virtual  Instrument

AU: thank you also for these recommended bibliography we will included also in our next publication. We read lots of article, and made a selection for this paper that we think, may be sufficient.
